# Phenotype-dependent habitat choice is too weak to cause assortative mating between *Drosophila melanogaster* strains differing in light sensitivity

Juan Ramón Peralta-Rincón[1]*, Fatima Zohra Aoulad[1], Antonio Prado[2], Pim Edelaar[1]

1 Department of Molecular Biology and Biochemical Engineering, Universidad Pablo de Olavide, Seville, Spain, 2 Department of Physiology, Anatomy and Cell Biology, Universidad Pablo de Olavide, Seville, Spain

* juan.peralta@imagencientifica.es

**Data Availability Statement:** All relevant data are within the manuscript and its Supporting Information files.

## Abstract

Matching habitat choice is gaining attention as a mechanism for maintaining biodiversity and driving speciation. It revolves around the idea that individuals select the habitat in which they perceive to obtain greater fitness based on a prior evaluation of their local performance across heterogeneous environments. This results in individuals with similar ecologically relevant traits converging to the same patches, and hence it could indirectly cause assortative mating when mating occurs in those patches. White-eyed mutants of *Drosophila* fruit flies have a series of disadvantages compared to wild type flies, including a poorer performance under bright light. It has been previously reported that, when given a choice, wild type *Drosophila simulans* preferred a brightly lit habitat while white-eyed mutants occupied a dimly lit one. This spatial segregation allowed the eye color polymorphism to be maintained for several generations, whereas normally it is quickly replaced by the wild type. Here we compare the habitat choice decisions of white-eyed and wild type flies in another species, *D. melanogaster*. We released groups of flies in a light gradient and recorded their departure and settlement behavior. Departure depended on sex and phenotype, but not on the light conditions of the release point. Settlement depended on sex, and on the interaction between phenotype and light conditions of the point of settlement. Nonetheless, simulations showed that this differential habitat use by the phenotypes would only cause a minimal degree of assortative mating in this species.

## Introduction

Reproductive isolation is one of the key requisites of speciation in sexual organisms: once a group of individuals within a population stops mating with the rest, they evolve independently [1]. Reproductive isolation is an extreme form of assortative mating resulting in separate gene pools, but weaker forms of assortative mating can still help to maintain genetic polymorphism [2–7]. Assortative mating and reproductive isolation might happen due to anatomical barriers [1,3,7], but they can also be caused by behavioral isolation [2–8].

**Funding:** This work was funded by grants from the Spanish Ministry of Economy and Competitiveness (grant refs. CGL2016-79483-P to P. E.; grant ref. BES-2017-081415 to J.R. P.-R.) with support from the European Regional Development Fund. https://www.mineco.gob.es/portal/site/mineco/ https://ec.europa.eu/regional_policy/en/funding/erdf/ The funders had no role in study design, data collection and analysis, decision to publish, or preparation of the manuscript.

**Competing interests:** The authors have declared that no competing interests exist.

Habitat choice is one such behavior which could cause assortative mating and even reproductive isolation [1–5,7,9–16]. Optimal ecological performance in heterogeneous environments is linked to the match between phenotype and environment [17]. When the environmental variability has a spatial component, habitat choice can increase this phenotype-environment match, and thereby increase fitness [13,17]. Furthermore, organisms that can evaluate an array of possible habitats and identify the environment that provides the best fit to their current phenotype, will likely improve their ecological performance [14,18–20]. This specific, phenotype-dependent form of habitat choice is referred to as "matching habitat choice" [11,19,21]. Some degree of assortative mating between individuals with similar ecologically-relevant traits may be expected, even in the absence of sexual selection [14,16,22], to the extent that mate choice occurs in the preferred habitat. Unfortunately, few experimental tests of this possibility have been undertaken [6,10,11,21].

Taxis, the movement towards or away from a stimulus, is perhaps one of the most illustrative examples of a behavior that can cause habitat choice. Nematodes [23–26] and various taxa of zooplankton [27,28] for instance, are known to use thermo and phototaxis respectively to find the most suitable environment. In taxic species, subpopulations with different behavior in response to the same stimulus are expected to reduce contact and acquire a certain degree of reproductive isolation in a heterogeneous habitat [15,29]. The behavior of fruit flies of the genus *Drosophila* is heavily influenced by light [30]. There exist at least two different known phototaxic responses in adults, namely "fast" and "slow" phototaxis [31–33]. Fast phototaxis occurs immediately after a fly is disturbed–by a sudden mechanical stimulus for example—and typically involves quick escape towards the light source, presumably displaying a "run for open space" response, observed in many flying insects. This response is conserved throughout the genus *Drosophila* and is probably the most studied kind of phototaxis [34–38]. Slow phototaxis happens in the absence of such a disturbance and can be thought of as light-dependent habitat choice. Unlike fast phototaxis, the valence and intensity of slow phototaxis is quite variable among species [30,39], phenotypes [31,40] and even (in some cases genetically identical) individuals [33,41,42].

Like any habitat preference [21], taxis can be due to genetic preferences (e.g. [15,29]), a preference for a familiar environment, or because of a local assessment of a better phenotype-environment match. Eye pigmentation affects how fruit flies perceive light and thus how they might react to it [40]. White-eyed mutants, caused by a recessive loss-of-function mutation in the gene *white*, are known to have reduced visual acuity and increased sensibility to light, receiving almost 20 times more light on their photoreceptors [43] compared to wild red-eyed flies [44,45]. It is therefore reasonable to expect that preferences for a darker or brighter habitat could depend on eye pigmentation.

Jones and Probert [2] reported that, in a mixed population of *Drosophila simulans*, white-eyed mutants were outcompeted in few generations by wild type flies in both strongly and dimly lit environments. This is because the mutation responsible for the white-eye phenotype also has an additional number of mildly impairing effects on the flies (see discussion) [46–48]. However, when given a mixture of strong and dimly lit environments, the allele causing the white-eye phenotype was maintained. They further showed that wild type flies preferred environments with much light, whereas white-eyed flies preferred dimly lit environments, and this habitat preference reduced competitive and sexual interactions. This result suggests that assortative mating and a degree of reproductive isolation occurred, consistent with the possibility that speciation could be driven by matching habitat choice [6,11,14,16,21].

In this study we test if eye color-related slow phototaxis could also result in phenotype-based spatial segregation and assortative mating in another species of fruit fly, *Drosophila melanogaster*, by comparing the distributions of white-eyed and wild type flies along a light gradient.

## Methods

### The testing arena

We established a light gradient by linearly arranging 5 cages with decreasing values of illuminance. Neighboring cages were connected by a 40mm diameter circular opening cut in each one of the wide lateral faces. The openings of the first and final cages that did not connect with any other cage were plugged with foam stoppers that provided an oxygen inlet.

As cages we used 165 x 105 x 105 mm (~1.5l) transparent plastic fauna cages (Savic company) with the bottom and side faces covered on the outside by a layer of black opaque adhesive vinyl. The top was covered by a rectangular piece of transparent acrylic (2 mm thick) and the openings between cages were lined with EVA foam to prevent the flies from escaping the arena (Fig 1A). A cardboard separator could be placed between two adjacent cages to block fly movement when needed. To prevent starvation and dehydration during the experiments, a petri dish with 10ml of sugary water ($10gl^{-1}$ white sugar) was placed inside each cage.

The cages where lit from above by regular linear fluorescent ceiling lamps (50 Hz AC) and the light intensity gradient was achieved by covering the transparent top face of each cage with a different number of layers of Neutral Density filter paper (Lee 0.3 ND filter). For the darkest cage, we covered the top with a layer of black adhesive vinyl which prevented any light from entering. This setup created a succession of spaces with decreasing illuminance values as seen in Fig 1B.

### Fly husbandry and data collection

We used outbred (genetically variable) populations of wild type and *white* ($w^{1118}$) *D. melanogaster* (creation of these lines described in [49]). All the stocks were reared on conventional wheat flour and agar medium under a 12h/12h light-dark cycle at 25°C. The experiments were performed with 2–3 days old adult flies collected from population cages, using both males and females (sex ratio close to 1). Since the flies were not separated by sex at any time, they were most likely all mated. Tested flies were not reused.

**a)** **b)**

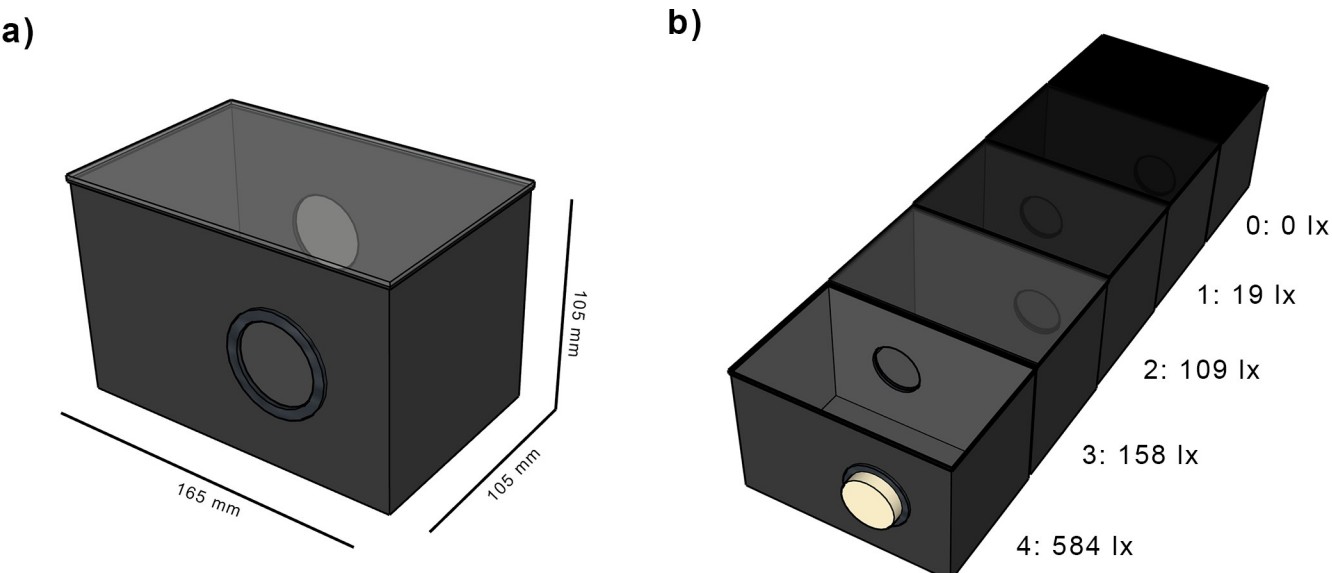

**Fig 1. Experimental setup.** Measurements for each cage module (a) and layout of all 5 cages (b). Cage 0 was covered by a layer of opaque vinyl; cages 1 to 3 were covered by 4, 2 and 1 layers of neutral density filter respectively. The resulting illuminance was measured at the bottom of the cage with a TeckoPlus digital lux meter. The openings on both ends of the arena were stopped by foam plugs.

Around 150±75 flies briefly anesthetized with $CO_2$ were released in one of the 5 cages of the setup in the late morning (around 5 hours after the lights were turned on). Following Heisenberg's [31] method for measuring slow phototaxis, the flies were allowed to roam all five cages of the arena for 20 hours under the same temperature and light-dark cycle as mentioned above. Experiments realized on the same day ran simultaneously in separate arenas. Temperature and relative humidity (65%) inside the setup were similar between cages.

Because dispersal decisions may depend on the initial point of release, release events were performed in each of the five cages, with two replicates per cage. In order to compare habitat choice between phenotypes, wild type *D. melanogaster* and white-eyed mutants were tested separately under the same conditions. Two replicates of each possible release cage for both strains yielded a total of 20 release events.

After 20 hours, the openings connecting neighboring cages were blocked and the flies anesthetized again by means of $CO_2$. We then counted and sexed the flies found inside each capture cage. When the cage of capture was the same as that of release, the flies were labeled as "resident", otherwise they were labeled as "disperser". To this data we added the information on phenotype and release cage, and an ID number for each release event. Once the flies were removed, the cages and petri dishes were wiped clean with a damp, soapy towel and dried before being reused in the following release event. Because cages and petri dishes were cleaned in no particular order, they were in effect shuffled when assembling arenas for new trials.

## Data analysis

We fitted binomial Generalized Linear Mixed Models (GLMMs) [50,51] using each individual fly's disperser status as the binary response variable. Sex, phenotype and either release or capture cage were used as possible predictors. For all models we added the release event ID number as a random effect in order to control for the possibility that flies were affected by unknown replicate-specific conditions. Dispersal is often divided into three separate stages: departure, transience and settlement [52], each of which might be affected by different variables. We studied the effect that phenotype has on habitat choice from two different perspectives. First, using release cage as a predictor, we tested whether or not the departure decision is affected by the light intensity inside the release cage. Second, using capture cage as a predictor, we tested whether or not the settlement decision is determined by the intensity of the light inside the capture cage. Care must be taken when interpreting these latter results, as a high disperser-to-resident ratio for a specific cage can be due to increased immigration or due to increased emigration. However, the combination of the two perspectives allows us to differentiate between these alternatives (see discussion).

For both of these perspectives, we fitted models with the following combinations of predictors: cage, phenotype+cage, phenotype*cage, sex+cage, sex+phenotype+cage, and sex+phenotype*cage. The phenotype*cage model ("*" means that both main effects plus their interaction are included as predictors in the model) is fitted to test our main hypothesis that habitat choice is phenotype-dependent, and sex is included to allow for any sexual dimorphism in dispersal. We selected the model with the best fit using the Akaike Information Criterion (AIC): a lower AIC indicates a stronger support for a given model.

## Simulation of expected degree of assortative mating

Based on the obtained results on dispersal decisions (see results), we simulated a scenario where equal numbers of wild type and white-eyed *D. melanogaster* of both sexes are released together in an arena similar to the one used in the experiment. Our simulation model, written in Python 3.8 (S1 Script) [53], was an individual-based simulation where each fly went through

the same four steps leading to mating: release, emigration, immigration and pairing. After the final step, we determined the degree of assortative mating of the population by calculating the proportion of same-phenotype pairs out of the total number of pairs (expected to be 0.5 on average in the absence of assortative mating, and 1.0 for fully assortative mating).

In the release step, 10 flies of every possible phenotype-sex combination were assigned to each one of the 5 cages for a total of 200 flies per replicate. After that, in the emigration step, flies decided whether or not to disperse based on their sex and phenotype following the probabilities obtained from the release cage perspective (see results). During the immigration step, disperser flies were assigned to new cages following the probabilities extracted from the capture cage perspective (see results).

The final step, pairing, involved each fly to be paired with a random individual of the opposite sex from the same cage. Each individual's choice was assumed to be independent from the rest so one mating event was recorded for each fly (both sexes were always present in each cage in the simulations we ran). The probability to mate and with whom was assumed to be independent of the phenotype, so we assumed no sexual selection (this might not be true in reality [46,54,55]) in order to only assess the effect of spatial distribution.

To determine the importance of the phenotype-based cage choice, we ran two control simulations with different immigration probabilities. In the first, the capture cage was chosen at random (i.e. no phenotype-dependent habitat choice) while in the other, white-eyed flies always chose cage 0 and wild type flies always chose cage 4 (i.e. habitat choice was completely phenotype-dependent). The remaining parameters were unchanged, and departure was always independent of eye-color phenotype. We performed 1000 iterations of each of the three simulation setups and compared the resulting proportions of same-phenotype pairs.

## Results

The raw data (Fig 2) shows that flies tend to disperse to cage 4 (the one with the most light), independent of the release cage. Nonetheless, the cage of release is often a local maximum in the distribution. In many cases, cage 3 seems to be avoided, attracting a lower number of flies than either of the neighboring cages.

### Release cage perspective

The best fitting model for the release cage perspective (Table 1) suggests that the probability to disperse from the release cage depends on the interaction between sex and phenotype (Fig 3), but not on the release cage itself. Starting in any given cage seems to have no effect on whether the fly departs or stays. In fact, among the fitted models, the one using release cage as the only explanatory variable has the worst fit to the data (Table 1).

### Capture cage perspective

When using capture cage as a predictor, the best fitting model contains sex and the interaction between phenotype and capture cage (Table 2, Fig 4). This indicates that the proportion of "resident" vs "disperser" flies captured in each cage depends on the phenotype of the flies.

### Simulation results

Our simulation suggests that the proportion of same-phenotype pairs expected to result from the observed mild phenotype-dependent settlement (Fig 4B) would be much closer to a scenario with random settlement than one where settlement is totally phenotype-dependent (Table 3, S1 Fig).

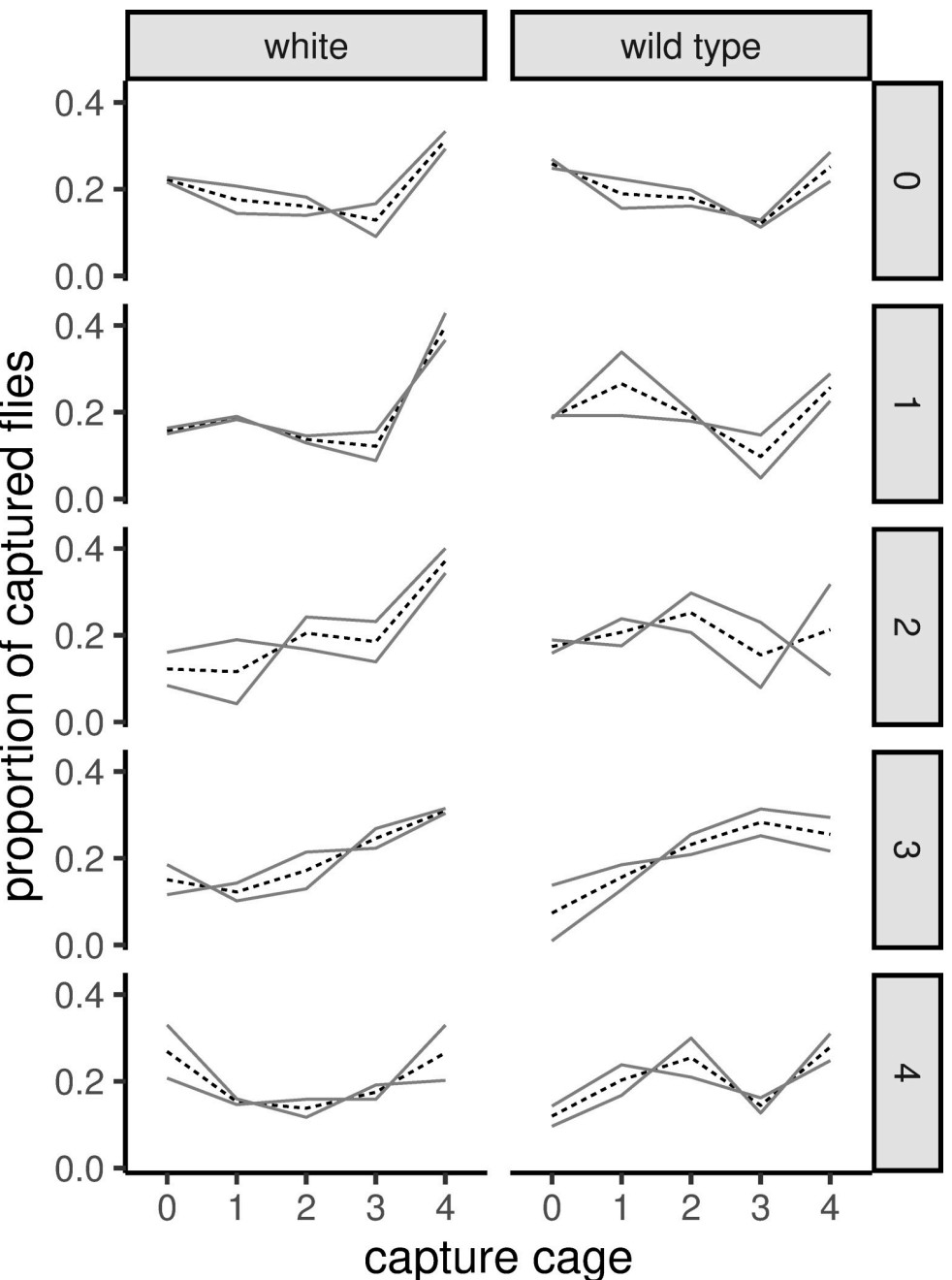

**Fig 2. Distribution of the captured flies depending on their phenotype and release cage.** Solid lines represent the two replicates and dashed lines show the mean of each treatment. Notice that there is a generalized upward trend towards cage 4. This is perhaps highlighted by the slight decrease often seen in cage 3 (except when flies are released in cage 3).

## Discussion

The environmental gradient design of our arena is derived from the binary choice between light/dark environments described in [2] for *D. simulans*, adding a number of intermediate options. We decided on this modification after failing to obtain obvious differences in spatial preference between *D. melanogaster* phenotypes in such a binary arena, as both red- and

**Table 1. Performance of the release cage perspective models.**

| Fitted model | ΔAIC |
|---|---:|
| **sex * phenotype** | **0.0** |
| sex + phenotype | 1.0 |
| sex + phenotype + release cage | 5.1 |
| sex | 6.1 |
| sex + release cage | 10.3 |
| sex + phenotype * release cage | 10.7 |
| phenotype | 26.9 |
| phenotype + release cage | 29.8 |
| release cage | 34.5 |

Relative performance of the fitted models compared to the best fit (in bold) based on their AIC value. The Akaike Information Criterion (AIC) is an estimator of the quality of a model; lower AIC values indicate a better fit of the model to the data

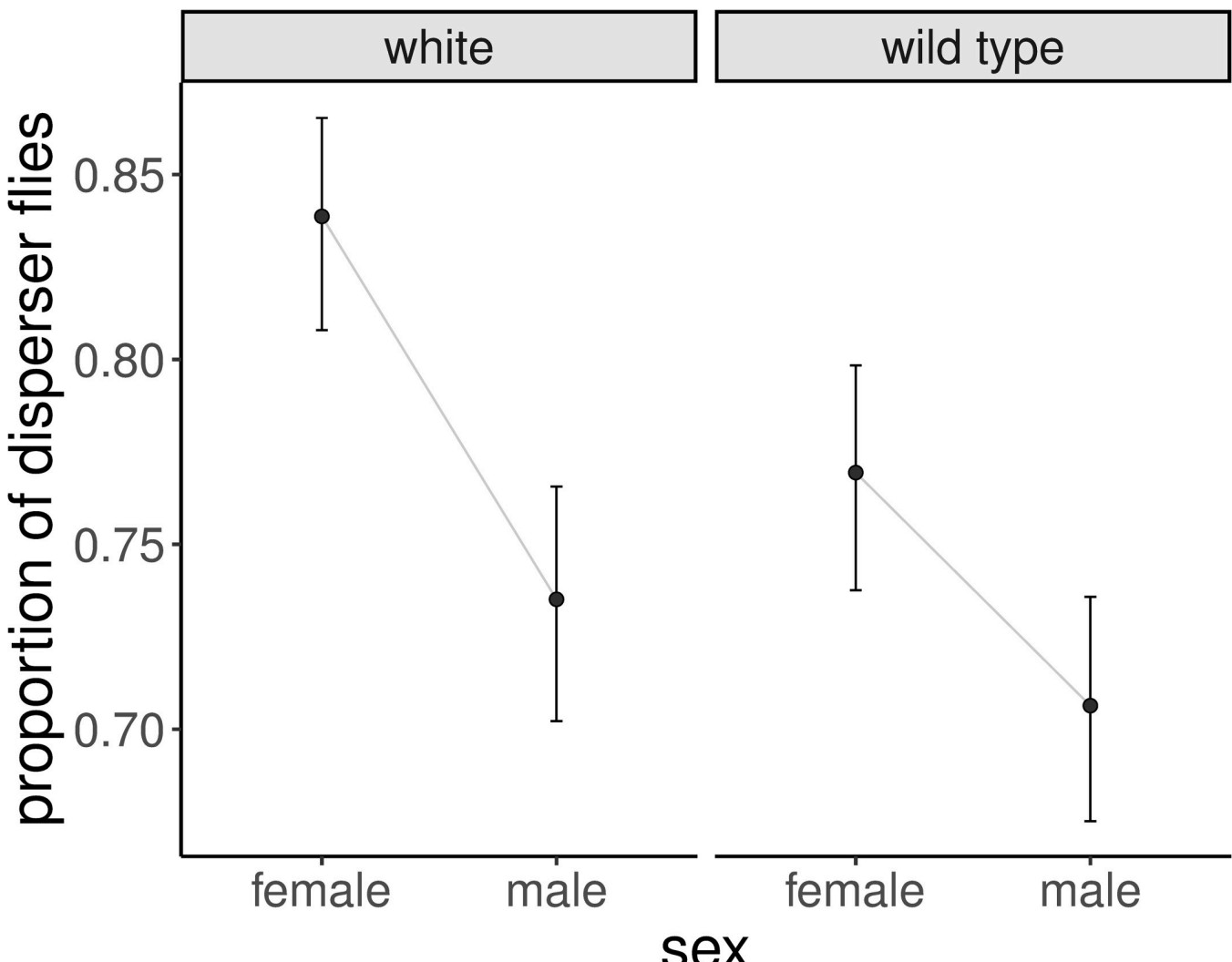

**Fig 3. Effects of the release cage perspective model.** Expected proportion of disperser flies for sex by phenotype as inferred from the best fitting model in Table 1. The error bars show the standard error of the mean. While flies are more likely to disperse if they are white-eyed or female, the model also suggests a synergy between both predictors. The average expected proportion of dispersers across all sex-phenotype combinations is 0.76 ± 0.06.

**Table 2. Performance of the capture cage perspective models.**

| Fitted model | ΔAIC |
|---|---:|
| **sex + phenotype * capture cage** | **0.0** |
| sex + phenotype + capture cage | 46.7 |
| sex + capture cage | 50.1 |
| phenotype + capture cage | 67.6 |
| capture cage | 70.2 |
| sex * phenotype | 244.8 |
| sex + phenotype | 245.8 |
| sex | 250.9 |
| phenotype | 271.7 |

Relative performance of the fitted models compared to the best fit (in bold) based on their AIC value. The Akaike Information Criterion (AIC) is an estimator of the quality of a model; lower AIC values indicate a better fit of the model to the data.

white-eyed flies clearly preferred the light environment (J.R. Peralta-Rincón, personal observation). This environmental gradient should provide greater resolution of phenotype-dependent habitat choice and may be closer to what flies would find in a natural situation. Indeed, we detected minor differences between phenotypes with respect to settlement (Fig 4B). As a downside, this arrangement conditions the accessibility to any given cage to the release cage. In order to correct for this effect, we compare the results across all possible release cages.

The fact that flies from all cages tend to disperse to cage 4 (Figs 2 and 4) can be interpreted as this cage (with the greatest amount of light) being the preferred habitat. But it is also possible that cage 4 is working as a "trap" where the bright light partially blinds and disorients the flies preventing them from exploring and choosing any of the other cages. [46] reports male white-eyed *Drosophila* having a worse courtship performance in daylight than in a dimly lit environment due to the bedazzlement caused by intense light. This coupled with poor spatial memory in white-eyed flies [56] could be biasing the distribution of the mutants towards cage 4. This is, however, less likely the case for red-eyed flies as they successfully navigate in much brighter daylight. Another possibility is that the poor food source and lack of adequate egg laying substrate enticed flies, particularly females (Figs 3 and 4) to move towards bright lighting expecting a way out of the set-up, similar to what happens during fast phototaxis. Alternatively, in the case of sheer erratic movement we would also expect flies to accumulate at the end cages, however this would still need to be combined with some sort of light dependence in order to explain the lower number of flies in cage 0.

We noticed that higher numbers of flies were often found at the release cages (Fig 2). This, coupled with the fact that about 24% of the flies across all sexes and phenotypes were captured in the release cage (Fig 3), suggests that a relatively large number of flies did not explore beyond the connection with the next cage. On the one hand, it is likely that some flies did in fact explore neighboring cages only to come back to the original one, thus inflating the proportion of non-dispersers. On the other hand, it is also possible that the design of the arena made fly mobility difficult to some extent.

When deciding on size and placement of the openings connecting neighboring cages, reaching a good compromise between facilitating fly mobility and maintaining separate, differently lit environments is the main goal. For this reason, we used an opening thirty times larger in area than the 40mm$^2$ described in [2]. This size allows more flies to walk and fly through while still keeping separate light environments.

There is also the possibility that the high number of non-dispersers was caused by impaired movement due to handling. However, if so, one would expect white-eyed flies to disperse less. This is because although biologically functional, white-eyed *Drosophila* suffer from a series of neurological disadvantages (other than poor vision) due to their malfunctioning *white* protein. These include reduced spatial memory [56], low copulation success [47] and progressive loco-motor deficiencies [48]. For this reason, care should be taken when interpreting results from behavioral tests using these (and other) mutants, and it seems reasonable to expect that handling would negatively affect white-eyed flies more. In contrast to that, white-eyed flies were more dispersive than wild type flies (Fig 3, Table 1), for which we do not have a good alternative hypothesis. It does suggest however, that handling effects did not reduce dispersal. We also found that females were more dispersive than males (Fig 3). Potentially, searching for a suitable egg-laying site might explain this [57], in line with our earlier explanation of why more flies were found in cage 4.

When looking at the proportions of dispersers in capture cages, we found that it differs between the cages (Fig 4). An increase in this proportion can be caused by either high emigration (a reduced number of residents) or high immigration (an increased number of dispersers). Although there is no evidence from the statistical models alone that supports a stronger influence of one process over the other, given that our previous results for the departure stage indicate that all cages have a similar probability of flies dispersing (Table 1; no effect of release cage), an increase in the proportion of dispersers in a cage is likely due to higher immigration. Both phenotypes showed a photopositive response, with cage 0 being the least attractive. Both also showed a noteworthy local minimum in cage 3. If we assume the hypothesis that cage 4 is acting as a trap, this valley can be explained by cage 4 "stealing" wandering flies by preventing

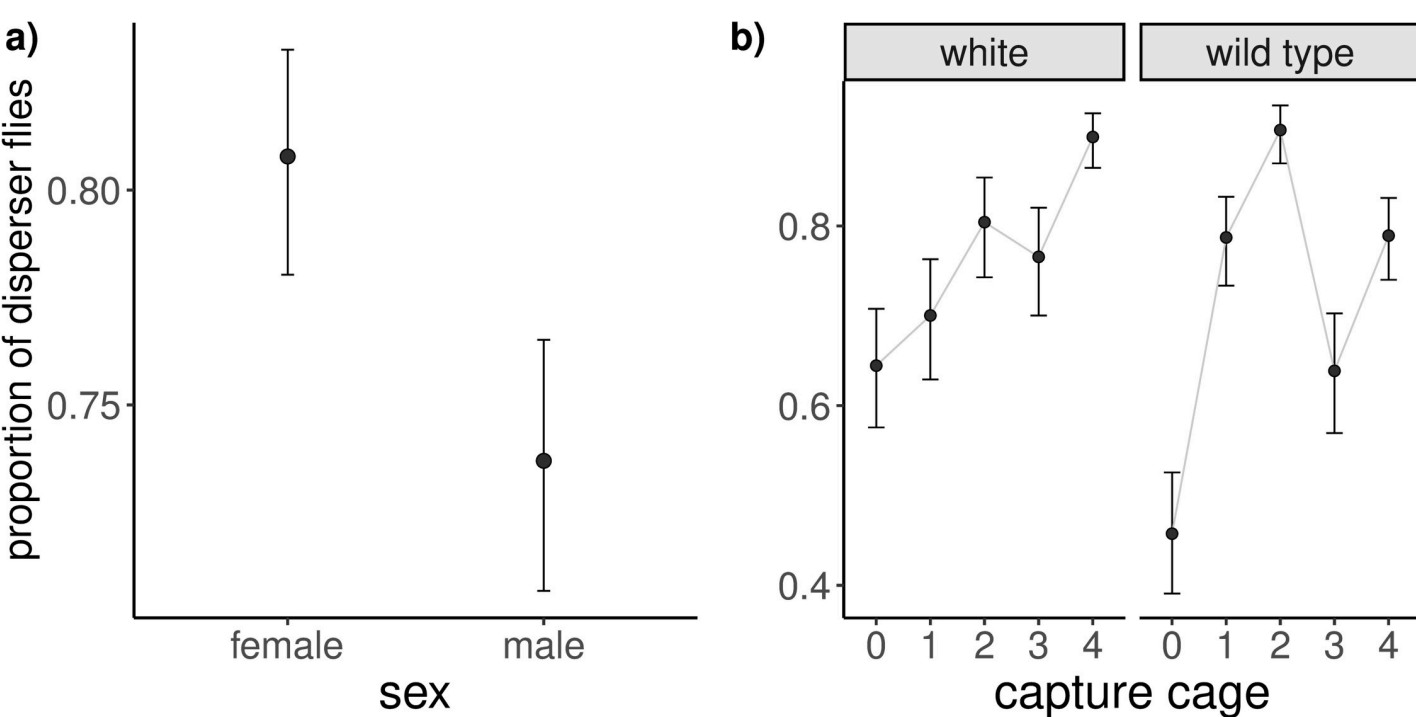

**Fig 4. Effects of the capture cage perspective model.** Expected proportion of disperser flies for sex (a) and phenotype by cage of capture (b) as obtained from the best fitting model in Table 2. The error bars show the standard error of the mean. Note for panel 4b that the pattern differs between wild type and white-eyed flies. For reference, the average expected proportion of dispersers across all flies is 0.76 (see Fig 3): for panel b, in the absence of habitat choice we would expect a similar proportion of dispersers in all cages.

**Table 3. Proportion of same-phenotype pairs for each simulation.**

| Cage choice | Proportion | SD | SE |
|---|---|---|---|
| Observed | 0.502 | 0.036 | 0.001 |
| Random | 0.497 | 0.037 | 0.001 |
| Forced | 0.844 | 0.034 | 0.001 |

Expected proportion of same-phenotype pairs for each simulated type of cage choice with standard deviation and standard error. Random habitat choice does not yield an expected mean score of 0.500 because emigration is still sex- and phenotype-dependent.

them from returning to cage 3. Another possible explanation is that flies cannot perceive a significant difference between cages 2 and 3 (with 109 and 158 lux respectively), which could reduce transit between those cages.

The best fitting model (Table 2) also suggests that the differences in proportions of dispersers between cages is phenotype-dependent: wild type flies seem more selective than white-eyed mutants as they show greater differences between cages (Fig 4B). These findings are in agreement with [42] who linked a serotonin deficiency caused by the mutated *white* gene to high behavioral variability in white-eyed *D. melanogaster*, which would add noise to any systematic habitat choice. It is also possible that the light powered by 50Hz alternating current (used to replicate [2]) was perceived as flickering and affected white-eyed and wild type flies differently. We understand that the responsiveness of both strains to pulses decreases for frequencies above 20 Hz [45], but nonetheless it might be useful to explore the effects of DC-powered lighting in further studies.

In spite of this statistically well-supported phenotype * release cage interaction (Table 2), the overall shape of the distribution is still rather similar for both phenotypes (Fig 4). While it is true that the absolute maxima are different for white-eyed and wild type flies (cages 4 and 2 respectively), these cages also attract a high proportion of flies of the other phenotype. In fact, our simulation predicts that the observed degree of phenotype-dependent habitat choice has only a minimal effect on the resulting assortative mating, being much closer to a random choice scenario than to a totally phenotype-dependent one (Table 3, S1 Fig). Hence, if both phenotypes were to be released together and homogeneously across cages in this experimental set-up, we would expect only a limited amount of spatial segregation between them, and extremely little assortative mating would result. In any case, it must be noted that the choice probabilities used for the females in this mating model come from the dispersal decisions of mated individuals and that the foraging and dispersal behavior of virgin females might be somewhat different [57].

Given the competitive disadvantage of the white allele, in the long run the population would probably lose the allele and become homozygous for the wild type, in contrast to the results of Jones & Probert [2]. Consequently, it is likely that for *D. melanogaster* phenotype-dependent habitat choice would not enable the eye color polymorphism to be maintained, as has been reported for *D. simulans*. Further experiments are necessary to determine to what extent phenotype-dependent habitat choice can maintain genetic polymorphism and even promote speciation, as has been suggested by theoretical studies [3,6,7,12,14,16,19,58] and a few empirical studies [2,59–61].

## Supporting information

**S1 Script. Code used for the simulation of the expected degree of assortative mating.** (PY)

**S1 Dataset. Data used for the statistical analyses.**
(CSV)

**S2 Dataset. Raw data obtained from the simulation experiments.**
(CSV)

**S1 Fig. Violin plot comparing the proportions of same-phenotype pairs obtained by the different types of cage choice.** Mean and standard deviation are shown.
(TIF)

## Acknowledgments

We thank Élio Sucena, Liliana Vieira and Tânia Paulo for kindly providing the outbred fly stocks used for the research as well as useful advice on their rearing. We would also like to thank Simone Santoro and Carlos Camacho for insightful discussion regarding the analysis of our data.

## Author Contributions

**Conceptualization:** Antonio Prado, Pim Edelaar.

**Formal analysis:** Juan Ramón Peralta-Rincón.

**Funding acquisition:** Pim Edelaar.

**Investigation:** Juan Ramón Peralta-Rincón, Fatima Zohra Aoulad.

**Methodology:** Juan Ramón Peralta-Rincón, Antonio Prado.

**Supervision:** Pim Edelaar.

**Writing – original draft:** Juan Ramón Peralta-Rincón, Pim Edelaar.

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
