## [Decision Letter · Decision Letter 0]

30 Jun 2020

PONE-D-20-15094

Phenotype-dependent habitat choice is too weak to cause assortative mating between Drosophila melanogaster strains differing in light sensitivity

PLOS ONE

Dear Dr. Peralta-Rincón,

Thank you for submitting your manuscript to PLOS ONE. After careful consideration, we feel that it has merit but does not fully meet PLOS ONE’s publication criteria as it currently stands. Therefore, we invite you to submit a revised version of the manuscript that addresses the points raised during the review process.

The MS is in generally good shape but we have some concerns about interpretation and presentation of the data.

Two reviewers were concerned about the design of the behavioral chamber and potential confounding factors. While, we will not ask you to preform extra experiments, but if you have auxiallary data (or cited works) that reflect on the issues raised by the reviewers, then please include them in the manuscript. Plos One has room for supplemental material. For instance but not exclusive to the reviers concerns about the dynamics of dispersal with different shading scheme, temp/humidity in chambers, and “End station effects” (like chamber 4). If extra data are not available, please discuss earnestly the weaknessess of the design/contraption and potential fixes, and add caveats about the interpretation of the data.

Secondly, the data are (if im interpreting your supplemental table correctly) from replicate experiments. Please present the raw numbers for each replicate, e.g. in fig 2 (and possibly also the mean), and evaluate the variance.

We look forward to receiving your revised manuscript.

Kind regards,

Arnar Palsson, Ph.D.

Academic Editor

PLOS ONE

Journal Requirements:

Additional Editor Comments (if provided):

The MS is in generally good shape but we have some concerns about interpretation and presentation of the data.

Two reviewers were concerned about the design of the behavioral chamber and potential confounding factors. While, we will not ask you to preform extra experiments, but if you have auxiallary data (or cited works) that reflect on the issues raised by the reviewers, then please include them in the manuscript. Plos One has room for supplemental material. For instance but not exclusive to the reviers concerns about the dynamics of dispersal with different shading scheme, temp/humidity in chambers, and “End station effects” (like chamber 4). If extra data are not available, please discuss earnestly the weaknessess of the design/contraption and potential fixes, and add caveats about the interpretation of the data.

Secondly, the data are (if im interpreting your supplemental table correctly) from replicate experiments. Please present the raw numbers for each replicate, e.g. in fig 2 (and possibly also the mean), and evaluate the variance.

Minor points.

Can you reword the start of the abstract, “Over the last few years” is rather casual.

I also suggest you tone down the language in the discussion on the potential evolutionary consequences of the observed patterns. For instance, line 360 “Consequently, it [is likely] that [in]

D. melanogaster phenotype-dependent habitat choice [would] not enable the eye color polymorphism...”

Reviewers' comments:

Reviewer's Responses to Questions

**Comments to the Author**

1. Is the manuscript technically sound, and do the data support the conclusions?

Reviewer #1: Partly

Reviewer #2: Partly

Reviewer #3: Partly

2. Has the statistical analysis been performed appropriately and rigorously? 

Reviewer #1: Yes

Reviewer #2: Yes

Reviewer #3: Yes

3. Have the authors made all data underlying the findings in their manuscript fully available?

Reviewer #1: Yes

Reviewer #2: Yes

Reviewer #3: Yes

4. Is the manuscript presented in an intelligible fashion and written in standard English?

Reviewer #1: Yes

Reviewer #2: No

Reviewer #3: Yes

5. Review Comments to the Author

Reviewer #1: This manuscript measures preference for light environment in wildtype and white-eyed Drosophila melanogaster, and found that (in contrast to D. simulans) there was no evidence that white-eyed flies avoided bright light. They then use empirical data on differences in habitat choice in a simulation model to determine whether assortative mating resulting from such choice could result in the maintenance of the white-eyed allele over evolutionary time. They conclude that this outcome is unlikely.

Overall I thought the experiment was well-described and straightforward. The statistical analysis seems appropriate, with one minor caveat discussed below. The combination of empirical data and simulation modelling is a strength, and helps to provide some broader relevance to the empirical results. However there are two weaknesses to the experimental set-up, which may or may not affect the robustness of the results. I leave it up to the editor to decide how important these issues are. Unfortunately I can't asses the simulation model directly since I don't work with python.

Firstly, since distribution across the cages was used as a measure of preference, I find it odd that no control conditions were included, e.g. all cages illuminated or in darkness. It seems like this would have been useful for potentially excluding/confirming some of the speculations in the discussion. Under constant light conditions it could be that more flies would tend to accumulate in the end cages (1 & 4) since they only have a single opening each, compared to the middle cages (2 & 3) with two openings each. This sort of effect in combination with a preference for higher illumination could explain why there were more flies of both phenotypes in cage 4 but few in cage 3.

Secondly, I would have liked some sort of confirmation that there are no other environmental differences between cages that could account for the preference, such as differences in temperature or humidity. This doesn't seem especially likely to me under standard indoor illumination, but it's still possible that e.g. cage 4 might have had a slightly higher temperature or higher humidity (due to increased evaporation in light conditions compared to dark conditions). If so, the habitat choice that was observed in the white-eyed flies might not be related to illumination after all.

Minor comments

Line 151: It would be good to include more information about what specific stocks were used, i.e. the origin of the wildtype lines.

Lines 169-172: It is of course entirely possible that a "resident" fly moved out of the release cage and then moved back in. It might be worth mentioning this. Could it be possible to estimate the number of true residents by looking at the excess flies in the home cage in figure 3?

Lines 192-197: Why was no sex*phenotype interaction included? Is there any a priori reason to expect that male and female behaviour is affected completely symetrically by the white-eye mutation?

Lines 235-238: Why are there no error bars in figure 2? It says in the methods section that there were 2 replicates in each condition.

Reviewer #2: The authors have utilized the popular model organism, Drosophila melanogaster to study the influence of phenotype-based habitat choice on spatial segregation. They compare the spatial segregation patterns of white-eyed and wild type flies in response to a light gradient. Their results show that departure from the cage of introduction was predicted mainly by sex and phenotype, whereas settlement behavior was predicted by sex, and an interaction between phenotype and light conditions. Contrary to their predictions, white eye mutants were more likely to disperse out of their starting cage than wild type flies. And females flies dispersed more than male flies. Additionally, the proportion of white eye flies recorded in the brightest cage was greater than the proportion of wildtype flies captured in the brightest cage. These findings are somewhat contradictory to their predictions and suggest that eye color-related differential habitat segregation is not strong in D. melanogaster (in this experimental setup).

Overall, the study has the potential to be an interesting and useful contribution to the speciation literature. However, a few changes to the writing, phrasing of the conclusion, and additions to the discussion are necessary for publication. I’ve provided detailed comments for every section below:

Abstract:

1) change “after a prior evaluation” to “based on a prior evaluation”

2) What does the phrase ecological traits mean for an individual? perhaps it refers to phenotypic traits that are suited for specific environments?

Introduction:

Line 63: change “will effectively” to “will likely”

Line 66: “To the extent that mating subsequently occurs in this preferred habitat, a degree of assortative mating between individuals with similar ecological traits will occur” – this statement is too definitive. Habitat preference can occur at various spatial scales, and this may influence assortative mating differently. Additionally, the nature and consequences of habitat preference likely differ by species. Please rephrase this statement to reflect that this is one hypothesis, and not a certainty.

Line 71: please specify what species/taxa this statement is referring to.

Line 100: This sentence is poorly phrased. It is unclear what the authors mean by “poorer biological preference”. Please elaborate on the measures of fitness used to assess white eye mutants.

Methods:

1) Change the subtitle “Fly stocks” to “Fly husbandry and data collection”. Then combine the two sections. Under this section, please add details about the room in which behavior was conducted: temperature, sources of ambient light, humidity, etc.

2) Sentence on lines 163-165 is poorly phrased. Please provide a clear sentence to highlight that the two strains were tested separately.

3) Please elaborate on why 20 hours was chosen.

4) Were the wt and mutant flies tested in succession on the same day? Were they tested in separate chambers? What time of the day were these experiments conducted? Was the time of day controlled for wt vs mutant flies? If the same chambers were reused, how were the chambers cleaned prior to the next experiment? Please provide these details.

5) What software was used for statistical modeling?

6) Please provide model output in the main text or supplement. What are the error bars? Please state if the error bars are 95% Cis or SEMs in every figure legend.

Results and discussion:

1) A local maximum in the release cage suggests that the test chamber design may not be conducive to dispersal. The hole between cages may be too small for them to find entry or exit. This is the principle behind some fly traps. If there is no motivation to find the hole, they can’t. Control experiments in chambers with many or larger entry points could resolve this issue. If the authors are unable to perform additional experiments, there needs to be a clear justification for why they used this kind of chamber design. They also need to include a detailed discussion of this low overall dispersal rate. They mentioned a sentence about chamber #4 acting as a “trap”, but this needs to be elaborated upon in the context of dispersal ability of fruit flies.

2) Authors need to discuss how the 20 hour light duration might affect fruit fly circadian clocks. The fly stocks were maintained in 12-hour light-dark chambers. An additional 8 hours of light during the experiment may influence fly physiology and hence affect their phototaxis/ dispersal propensities. I’m also concerned that the two different stains may be differently affected by this circadian stress. Please discuss.

3) Controls: The authors cite literature that shows white eye mutants are weaker and poor at spatial memory. However, in their findings, white eye mutants appear to be more likely to disperse than wild type flies. Are the mutants in-fact better dispersers or is this finding an artifact of the experimental design? A good control experiment would be to look at dispersal patterns in a different assay chamber and in the absence of light gradients.

4) Must tie the results to the hypothesis better. This is an issue throughout the manuscript. For e.g. it is not sufficient to state that there is a phenotype by light condition interaction. Please elaborate on the direction of the effect and how it relates to white eye vs. wild type flies.

5) Line 291-292: “we managed to detect minor differences between phenotypes with respect to settlement”. Please remove the words “managed to”. Rephrase to “ We detected minor…”

6) Please rephrase this sentence to make the meaning clearer: “As a downside, it also means that access to any given cage is conditioned by the release cage, but we took this into account statistically”.

Reviewer #3: The authors of this manuscript utilize a behavioral assay to determine if phenotype-dependent habitat choice could lead to assortative mating. The question itself is interesting, and the experimental method that they use is interesting and novel. However, the manuscript lacks clarity about the experimental design and data that significantly hinders an understanding of their findings.

Perhaps the biggest criticism stems from Figure 2 and the depiction of the raw data. The authors perform 2 replicate trials, releasing 150 +/- 75 flies at a time of either wild-type or white-eyed phenotype in one of 5 cages. In figure 2 they plot the raw numbers from these experimental trials, which will then be used to fit binomial Generalized Linear Mixed Models (GLMM) plus a later model to assess the likelihood of assortative mating. Therefore, if the authors have not collected enough replicates to fully explore the natural variation that occurs during dispersal, their model will be heavily biased. As such, it is important to reassure readers of the quality of the collected data. To do so, the authors should not plot an aggregate of all data regardless of phenotype or trial (as seems to be the case in figure 2), but rather the raw numbers for each of the scenarios. In other words, each row of figure 2 should have four lines – each line would represent the numbers attained from a single trial, and there were two trials per the two phenotypes. Alternatively, if they had more than two replicates, then they could plot the data for each phenotype with some indication of variability (i.e., S.E.M). Either representation of the data will allow readers to better asses the quality of the data obtained and how much dispersal may have varied across the two replicates. Additionally, most readers will likely not be familiar with GLMM methods, and providing these raw numbers will aid their understanding.

Below I list some other major points of note:

1. Authors provide a nice introduction to the topic and its importance. I very much enjoyed reading it.

2. I also liked the use of the two models, from the release cage perspective and the capture cage perspective. That these were both implemented by the authors suggests a nuanced understanding of dispersal.

3. Figure legends are spare, often slowing interpretation of the figures.

4. Figure 2 -- Is the slight decrease seen in cage 3 actually significant? What about the slight increase of flies present in cage 4? Did the authors clean the cages between trials to remove any chemosensory cues?

5. Figure 2 – I believe the y-axis should read “proportion of captured flies”, not percentage.

6. Were the females tested virgins or already mated? The methods section suggestions that they were collected at the same time as males, which likely means that they were already mated. Is it possible that virgin and mated females have different light preferences?

7. It is unclear what the “*” means for the phenotype*cage interaction (compared to the phenotype + cage interaction). It is likely standard nomenclature, but readers would benefit from an explanation (since most will be unfamiliar with GLMM methods).

8. Figure 4 – It is unclear if this figure plots the results of the raw data or data resulting from the GLMM. If it plots data resulting from the GLMM, then please provide the raw numbers actually obtained from the experiments (essentially, what I propose plotting in figure 2). Please also include an explanation as to why 0.77 would be expected when there is an absence of habitat choice and why the proportion of disperser flies in both panels do not approximately add up to one.

9. The modeling results section lacks description of the model nor does it have an accompanying figure or table. In general, I found this section of the manuscript the most lacking and hard to interpret. As a result, I do not feel that I can trust the results of the model. This model is the entirety of the evidence that the small differences in dispersal seen by the authors would not cause assortative mating. More time should be spent explaining it and interpreting its results.

6. PLOS authors have the option to publish the peer review history of their article (what does this mean?). If published, this will include your full peer review and any attached files.

Reviewer #1: No

Reviewer #2: No

Reviewer #3: **Yes: **Sweta Agrawal

---

## [Author Response · Author response to Decision Letter 0]

14 Aug 2020

Editor

COMMENT: Two reviewers were concerned about the design of the behavioral chamber and potential confounding factors. While, we will not ask you to preform extra experiments, but if you have auxiallary data (or cited works) that reflect on the issues raised by the reviewers, then please include them in the manuscript. Plos One has room for supplemental material. For instance but not exclusive to the reviers concerns about the dynamics of dispersal with different shading scheme, temp/humidity in chambers, and “End station effects” (like chamber 4). If extra data are not available, please discuss earnestly the weaknessess of the design/contraption and potential fixes, and add caveats about the interpretation of the data.

REPLY: We added the requested details of the conditions inside the behavioral chamber (humidity, temperature, light/dark cycles) and discussed the reviewers’ concerns at the best of our ability (see below).

C: Secondly, the data are (if im interpreting your supplemental table correctly) from replicate experiments. Please present the raw numbers for each replicate, e.g. in fig 2 (and possibly also the mean), and evaluate the variance.

R: Fig 2 now shows the whole data, with replicates separated and the mean.

C: Minor points.

Can you reword the start of the abstract, “Over the last few years” is rather casual.

R: Changed. It now reads “Matching habitat choice is gaining attention…” (line 26)

C: I also suggest you tone down the language in the discussion on the potential evolutionary consequences of the observed patterns. For instance, line 360 “Consequently, it [is likely] that [in]

D. melanogaster phenotype-dependent habitat choice [would] not enable the eye color polymorphism...”

R: We changed the suggested phrase and some other details of the discussion. (line 400)

Reviewer 1 

C: This manuscript measures preference for light environment in wildtype and white-eyed Drosophila melanogaster, and found that (in contrast to D. simulans) there was no evidence that white-eyed flies avoided bright light. They then use empirical data on differences in habitat choice in a simulation model to determine whether assortative mating resulting from such choice could result in the maintenance of the white-eyed allele over evolutionary time. They conclude that this outcome is unlikely.

Overall I thought the experiment was well-described and straightforward. The statistical analysis seems appropriate, with one minor caveat discussed below. The combination of empirical data and simulation modelling is a strength, and helps to provide some broader relevance to the empirical results. However there are two weaknesses to the experimental set-up, which may or may not affect the robustness of the results. I leave it up to the editor to decide how important these issues are. Unfortunately I can't asses the simulation model directly since I don't work with python.

Firstly, since distribution across the cages was used as a measure of preference, I find it odd that no control conditions were included, e.g. all cages illuminated or in darkness. It seems like this would have been useful for potentially excluding/confirming some of the speculations in the discussion. Under constant light conditions it could be that more flies would tend to accumulate in the end cages (1 & 4) since they only have a single opening each, compared to the middle cages (2 & 3) with two openings each. This sort of effect in combination with a preference for higher illumination could explain why there were more flies of both phenotypes in cage 4 but few in cage 3.

R: We acknowledge this limitation in our design and added the possibility of the “end station” effect to the discussion (line 328). 

Since the aim of the experiment was not so much to test the dispersal behavior and habitat choice of the flies by themselves but whether they were different in the two strains, we only compared these to each other, and not to additional control situations which would allow additional inference. Unfortunately, we had limited time and manpower during the experimentation phase, and this affected the number of replicates.

C: Secondly, I would have liked some sort of confirmation that there are no other environmental differences between cages that could account for the preference, such as differences in temperature or humidity. This doesn't seem especially likely to me under standard indoor illumination, but it's still possible that e.g. cage 4 might have had a slightly higher temperature or higher humidity (due to increased evaporation in light conditions compared to dark conditions). If so, the habitat choice that was observed in the white-eyed flies might not be related to illumination after all.

R: To determine this, we recreated the setup using the same cages and measured the temperature, relative humidity and evaporation inside each cage. Together with the room temperature, they were all within a 0.3C range with no gradients. RH and evaporation are also consistent between cages. This information has been added to the methods section. 

C: Minor comments

Line 151: It would be good to include more information about what specific stocks were used, i.e. the origin of the wildtype lines. 

R: The article cited in line 153 (49 in ref list) explains how the lines were established.

Lines 169-172: It is of course entirely possible that a "resident" fly moved out of the release cage and then moved back in. It might be worth mentioning this. Could it be possible to estimate the number of true residents by looking at the excess flies in the home cage in figure 3?

R: The simplest way to estimate this while maintaining release cage independence, assumes that the estimated mean proportion of disperser flies (0.76, Fig 3 legend) needs to be increased by a 25% (5 cages instead of 4). From this, we obtain that about 19% of the total flies disperse and go back to the release cage. This would mean that the proportion of true residents inside the release cage should be expected to be around 0.21 (5% of the total number of flies released). The problem with this estimation is that it is entirely based on conjecture.

 We could also use the immigration probabilities obtained from the model to find a true-resident proportion for each, but that would violate the cage-independent emigration values we assumed to obtain the immigration proportions in the first place. 

It is true nevertheless, that it is likely that an important number of the flies found in the release cage have explored and came back. This is now mentioned in the text. (line 335)

C: Lines 192-197: Why was no sex*phenotype interaction included? Is there any a priori reason to expect that male and female behaviour is affected completely symetrically by the white-eye mutation?

R: It was not part of our question initially; we were interested in seeing whether or not the cage*phenotype interaction had had significant effects. However, we have added a sex*phenotype model to the comparison (changes in tables 1 and 2). Sex*phenotype is now the best fit for the release perspective, just above sex + phenotype. The best fit for the capture perspective remains unchanged. 

We did not add a (sex*phenotype + cage*phenotype) model because the number of replicates we have provide too little statistical power to reliably estimate that many parameters. However, the fit of that model is still worse than that of the previously described ones. 

C: Lines 235-238: Why are there no error bars in figure 2? It says in the methods section that there were 2 replicates in each condition.

R: Also in response to the editor, Fig 2 now shows the whole data, with replicates separated and the mean. 

Reviewer 2 

C: The authors have utilized the popular model organism, Drosophila melanogaster to study the influence of phenotype-based habitat choice on spatial segregation. They compare the spatial segregation patterns of white-eyed and wild type flies in response to a light gradient. Their results show that departure from the cage of introduction was predicted mainly by sex and phenotype, whereas settlement behavior was predicted by sex, and an interaction between phenotype and light conditions. Contrary to their predictions, white eye mutants were more likely to disperse out of their starting cage than wild type flies. And females flies dispersed more than male flies. Additionally, the proportion of white eye flies recorded in the brightest cage was greater than the proportion of wildtype flies captured in the brightest cage. These findings are somewhat contradictory to their predictions and suggest that eye color-related differential habitat segregation is not strong in D. melanogaster (in this experimental setup).

Overall, the study has the potential to be an interesting and useful contribution to the speciation literature. However, a few changes to the writing, phrasing of the conclusion, and additions to the discussion are necessary for publication. I’ve provided detailed comments for every section below:

Abstract:

1) change “after a prior evaluation” to “based on a prior evaluation”

R: Changed. (line 28)

C: 2) What does the phrase ecological traits mean for an individual? perhaps it refers to phenotypic traits that are suited for specific environments?

R: Exactly, we understand ecological traits as any individual phenotypic feature that may influence an organism’s interactions with its environment. It was nevertheless modified to “ecologically relevant traits”. (line 30)

C: Introduction:

Line 63: change “will effectively” to “will likely”

R: Changed.

C: Line 66: “To the extent that mating subsequently occurs in this preferred habitat, a degree of assortative mating between individuals with similar ecological traits will occur” – this statement is too definitive. Habitat preference can occur at various spatial scales, and this may influence assortative mating differently. Additionally, the nature and consequences of habitat preference likely differ by species. Please rephrase this statement to reflect that this is one hypothesis, and not a certainty.

R: Rephrased to “Some degree of assortative mating between individuals with similar ecologically-relevant traits may be expected, even in the absence of sexual selection, to the extent that mate choice occurs in the preferred habitat” (line 66)

C: Line 71: please specify what species/taxa this statement is referring to.

R: Examples (thermotaxis in nematodes and phototaxis in zooplankton) are provided (line 73). 

C: Line 100: This sentence is poorly phrased. It is unclear what the authors mean by “poorer biological preference”. Please elaborate on the measures of fitness used to assess white eye mutants.

R: The text reads “poorer biological performance”, not “preference”. It is unclear to us whether this is a misreading (the reviewer found the term understandably alien) or a miswriting issue. In any case, we rephrased and now describe the white-eye mutation as “a pleiotropic trait with a number of mildly impairing effects”. (line 103) 

C: Methods:

1) Change the subtitle “Fly stocks” to “Fly husbandry and data collection”. Then combine the two sections. Under this section, please add details about the room in which behavior was conducted: temperature, sources of ambient light, humidity, etc.

R: Sections merged. Temperature, humidity and light-cycle details are now indicated. The source and amount of ambient light can be found in the “testing arena” section and figure 1b. 

C: 2) Sentence on lines 163-165 is poorly phrased. Please provide a clear sentence to highlight that the two strains were tested separately.

R: The previous sentence: “In order to compare habitat choice between phenotypes, separate experiments under the same conditions were carried out with wild type D. melanogaster flies and white-eyed mutants, for a total of 20 release events.” 

Is now rephrased to:

“In order to compare habitat choice between phenotypes, wild type D. melanogaster and white-eyed mutants were tested separately under the same conditions. Two replicates of each possible release cage for both strains yielded a total of 20 release events”. (line 169)

C: 3) Please elaborate on why 20 hours was chosen.

R: This has been made clearer in the text. This amount of time follows a classical article on measuring slow phototaxis [30 in ref list] and, as a practical advantage, allowed us to collect the flies and clean the setup in time to start the next experiment during our regular working hours (line 163). 

C: 4) Were the wt and mutant flies tested in succession on the same day? Were they tested in separate chambers? What time of the day were these experiments conducted? Was the time of day controlled for wt vs mutant flies? If the same chambers were reused, how were the chambers cleaned prior to the next experiment? Please provide these details. 

R: Each day, we started up to 4 parallel experiments (depending on fly availability), in the late morning (about 5 hours after the lights were automatically switched on in the lab). The strains were tested separately. The chambers were wiped clean between uses with a damp, soapy towel. These details are now in the text (lines 162, 165 and 179, respectively). 

C: 5) What software was used for statistical modeling?

R: The model was written from scratch using version 3.8 of the language Python. We clarified the version in the text (line 215). 

C: 6) Please provide model output in the main text or supplement. What are the error bars? Please state if the error bars are 95% Cis or SEMs in every figure legend.

R: Full model output and visualization are now available. The error bars in figures 3 and 4 are SEMs: this is now indicated in the caption. 

C: Results and discussion:

1) A local maximum in the release cage suggests that the test chamber design may not be conducive to dispersal. The hole between cages may be too small for them to find entry or exit. This is the principle behind some fly traps. If there is no motivation to find the hole, they can’t. Control experiments in chambers with many or larger entry points could resolve this issue. If the authors are unable to perform additional experiments, there needs to be a clear justification for why they used this kind of chamber design. They also need to include a detailed discussion of this low overall dispersal rate. They mentioned a sentence about chamber #4 acting as a “trap”, but this needs to be elaborated upon in the context of dispersal ability of fruit flies.

R: One of our main goals was to test whether or not the light condition was strong enough a motivation to elicit exploration and dispersal in fruit flies. 

The connecting slit described in Jones & Probert [ref 2] was a 20mm x 2mm. That seemed a bit restrictive to us, so we tested various hole sizes. A 40mm diameter circle was the biggest opening that didn’t compromise the plastic cage structure nor let too much light bleed inside the darker cages. In the data we see flies of both phenotypes and sexes use the holes in every possible direction. When a stronger motivation (such as food for starved flies) was provided, all flies could easily navigate through an opening this size (personal observation). 

This is now discussed in the text (lines 338-348). 

C: 2) Authors need to discuss how the 20 hour light duration might affect fruit fly circadian clocks. The fly stocks were maintained in 12-hour light-dark chambers. An additional 8 hours of light during the experiment may influence fly physiology and hence affect their phototaxis/ dispersal propensities. I’m also concerned that the two different stains may be differently affected by this circadian stress. Please discuss.

R: During the experiment, the flies were under the same light-dark cycle. This was not explicitly mentioned in the methods section and the expression “allowed to roam undisturbed” might have been misleading. This was changed (line 164). 

C: 3) Controls: The authors cite literature that shows white eye mutants are weaker and poor at spatial memory. However, in their findings, white eye mutants appear to be more likely to disperse than wild type flies. Are the mutants in-fact better dispersers or is this finding an artifact of the experimental design? A good control experiment would be to look at dispersal patterns in a different assay chamber and in the absence of light gradients. 

R: This is an unexpected result, for which we have no explanation. We agree that more experiments could help determine whether mutants are better dispersers in any way, but this is beyond the scope of this article. 

C: 4) Must tie the results to the hypothesis better. This is an issue throughout the manuscript. For e.g. it is not sufficient to state that there is a phenotype by light condition interaction. Please elaborate on the direction of the effect and how it relates to white eye vs. wild type flies.

R: Various parts of the discussion were modified to improve the continuity with the results (e.g. in lines 325, 332, 332 and 395). 

C: 5) Line 291-292: “we managed to detect minor differences between phenotypes with respect to settlement”. Please remove the words “managed to”. Rephrase to “We detected minor…”

R: Changed (line 310).

C: 6) Please rephrase this sentence to make the meaning clearer: “As a downside, it also means that access to any given cage is conditioned by the release cage, but we took this into account statistically”.

R: Rephrased to “As a downside, this arrangement conditions accessibility to any given cage to the release cage. We mitigate this effect statistically by comparing the results across all possible release cages” (line 311).

Reviewer 3

C: The authors of this manuscript utilize a behavioral assay to determine if phenotype-dependent habitat choice could lead to assortative mating. The question itself is interesting, and the experimental method that they use is interesting and novel. However, the manuscript lacks clarity about the experimental design and data that significantly hinders an understanding of their findings.

Perhaps the biggest criticism stems from Figure 2 and the depiction of the raw data. The authors perform 2 replicate trials, releasing 150 +/- 75 flies at a time of either wild-type or white-eyed phenotype in one of 5 cages. In figure 2 they plot the raw numbers from these experimental trials, which will then be used to fit binomial Generalized Linear Mixed Models (GLMM) plus a later model to assess the likelihood of assortative mating. Therefore, if the authors have not collected enough replicates to fully explore the natural variation that occurs during dispersal, their model will be heavily biased. As such, it is important to reassure readers of the quality of the collected data. To do so, the authors should not plot an aggregate of all data regardless of phenotype or trial (as seems to be the case in figure 2), but rather the raw numbers for each of the scenarios. In other words, each row of figure 2 should have four lines – each line would represent the numbers attained from a single trial, and there were two trials per the two phenotypes. Alternatively, if they had more than two replicates, then they could plot the data for each phenotype with some indication of variability (i.e., S.E.M). Either representation of the data will allow readers to better asses the quality of the data obtained and how much dispersal may have varied across the two replicates. Additionally, most readers will likely not be familiar with GLMM methods, and providing these raw numbers will aid their understanding.

R: Figure 2 has been modified to show each replicate separately, grouped by release cage and phenotype and the mean between replicates. We are aware that the low replicate count gives our model a limited statistical power, but we don’t believe the observed patterns are systematically biased in one direction. 

C: Below I list some other major points of note:

1. Authors provide a nice introduction to the topic and its importance. I very much enjoyed reading it.

R: Thank you!

C: 2. I also liked the use of the two models, from the release cage perspective and the capture cage perspective. That these were both implemented by the authors suggests a nuanced understanding of dispersal.

R: We are glad the reviewer found our approach interesting and thank her for the compliments.

C: 3. Figure legends are spare, often slowing interpretation of the figures.

R: We rewrote and expanded the legends of Figs. 2 to 4 and modified that of Fig 1 to improve clarity. 

C: 4. Figure 2 -- Is the slight decrease seen in cage 3 actually significant? What about the slight increase of flies present in cage 4? Did the authors clean the cages between trials to remove any chemosensory cues?

R: We did not test for specific cage effects, just if there was a general effect of cage (see Tables 1 and 2). However, Fig. 4B suggests that the pattern for cage 3 was indeed different from the others, which is why we spend some space on discussing this.

All cages and lids were wiped clean between experiments. We did not track individual cages while cleaning and, since the only difference between them was the light filter placed on the lid, they were likely rearranged between experiments. We added this to the manuscript (line 179)

C: 5. Figure 2 – I believe the y-axis should read “proportion of captured flies”, not percentage.

R: Changed.

C: 6. Were the females tested virgins or already mated? The methods section suggestions that they were collected at the same time as males, which likely means that they were already mated. Is it possible that virgin and mated females have different light preferences?

R: They were already mated; this is now explicit in the text. (line 155)

Mated females have an oviposition-site searching behavior that is absent in virgins [ref 57 on the reference list] and males, so different choice probabilities for females are indeed likely. This is noted in the text (line 395) 

C: 7. It is unclear what the “*” means for the phenotype*cage interaction (compared to the phenotype + cage interaction). It is likely standard nomenclature, but readers would benefit from an explanation (since most will be unfamiliar with GLMM methods).

R: It means phenotype and cage are used separately as predictors along with their interaction. We clarified this in the text (line 203). 

C: 8. Figure 4 – It is unclear if this figure plots the results of the raw data or data resulting from the GLMM. If it plots data resulting from the GLMM, then please provide the raw numbers actually obtained from the experiments (essentially, what I propose plotting in figure 2). Please also include an explanation as to why 0.77 would be expected when there is an absence of habitat choice and why the proportion of disperser flies in both panels do not approximately add up to one.

R: Proportions do not add up to one because each plotted point is a separate binomial probability: the point on plot 4a at (female, 0.81) indicates that 81 % of female flies are expected to disperse (0.19 are expected to remain in the release cage). 

0.77 (0.76 after modifying the release cage model) is the average proportion of disperser flies obtained from the release cage model. Since this was not mentioned before, this number seems to come out of nowhere and we understand the confusion. We modified the legends of Figs. 3 and 4 to explain this and to clarify that the plotted data comes from the GLMM output of the chosen models. 

Fig. 2 has also been modified to present the raw data in a clearer manner. 

C: 9. The modeling results section lacks description of the model nor does it have an accompanying figure or table. In general, I found this section of the manuscript the most lacking and hard to interpret. As a result, I do not feel that I can trust the results of the model. This model is the entirety of the evidence that the small differences in dispersal seen by the authors would not cause assortative mating. More time should be spent explaining it and interpreting its results.

R: The modeling methods and results sections are now hopefully improved and some discrepancies that made them feel disjointed have been corrected. We also provide the full output of the model as well as a visualization of the data (s3 and s4) and a table (table 3) with the assortative mating scores.

---

## [Decision Letter · Decision Letter 1]

22 Sep 2020

PONE-D-20-15094R1

Phenotype-dependent habitat choice is too weak to cause assortative mating between Drosophila melanogaster strains differing in light sensitivity

PLOS ONE

Dear Dr. Peralta-Rincón,

Thank you for submitting your manuscript to PLOS ONE. After careful consideration, we feel that it has merit but does not fully meet PLOS ONE’s publication criteria as it currently stands. Therefore, we invite you to submit a revised version of the manuscript that addresses the points raised during the review process.

The manuscript is greatly improved. I just have a couple of minor suggestions.

I also recommend that you read over the language carefully, as there might have been some other bloopers that I missed. Mind in particular the figure legends and table headings.

Minor

“This is because the white-eye mutation is a pleiotropic trait with a number of mildly impairing effects on the ...”

Actually the mutation is not a trait! Rephrase, and explain which traits are affected by mutations in white?

delta AIC in table - define as a footnote.

“ (heading level 2)”

Line 268

“Solid lines represent TWO replicates”

Line 280

Reword. “only explanatory variable fits the worst (Table 1)”

Defined “SEM”

Line 319

Reword “and very different form one where” and “choice is maximally phenotype-dependent”

Table 3.

Make title more transparent

“Assortative mating scores for each simulation.”

Line 337

Expand, names of authors “(personal observation)”

Line 342

Still poor wording “We mitigate this effect statistically by...”

We look forward to receiving your revised manuscript.

Kind regards,

Arnar Palsson, Ph.D.

Academic Editor

PLOS ONE

Additional Editor Comments (if provided):

The manuscript is greatly improved. I just have a couple of minor suggestions.

I also recommend that you read over the language carefully, as there might have been some other bloopers that I missed. Mind in particular the figure legends and table headings.

Minor

“This is because the white-eye mutation is a pleiotropic trait with a number of mildly impairing effects on the ...”

Actually the mutation is not a trait! Rephrase, and explain which traits are affected by mutations in white?

delta AIC in table - define as a footnote.

“ (heading level 2)”

Line 268

“Solid lines represent TWO replicates”

Line 280

Reword. “only explanatory variable fits the worst (Table 1)”

Defined “SEM”

Line 319

Reword “and very different form one where” and “choice is maximally phenotype-dependent”

Table 3.

Make title more transparent

“Assortative mating scores for each simulation.”

Line 337

Expand, names of authors “(personal observation)”

Line 342

Still poor wording “We mitigate this effect statistically by...”

Reviewers' comments:

Reviewer's Responses to Questions

**Comments to the Author**

1. If the authors have adequately addressed your comments raised in a previous round of review and you feel that this manuscript is now acceptable for publication, you may indicate that here to bypass the “Comments to the Author” section, enter your conflict of interest statement in the “Confidential to Editor” section, and submit your "Accept" recommendation.

Reviewer #1: All comments have been addressed

Reviewer #2: All comments have been addressed

Reviewer #3: All comments have been addressed

2. Is the manuscript technically sound, and do the data support the conclusions?

Reviewer #1: Yes

Reviewer #2: Yes

Reviewer #3: Yes

3. Has the statistical analysis been performed appropriately and rigorously? 

Reviewer #1: Yes

Reviewer #2: Yes

Reviewer #3: Yes

4. Have the authors made all data underlying the findings in their manuscript fully available?

Reviewer #1: Yes

Reviewer #2: Yes

Reviewer #3: Yes

5. Is the manuscript presented in an intelligible fashion and written in standard English?

Reviewer #1: Yes

Reviewer #2: Yes

Reviewer #3: Yes

6. Review Comments to the Author

Reviewer #1: (No Response)

Reviewer #2: The authors have done a good job in cleaning up language and providing clarifications. Here are two comments that need to be addressed:

typo on line 341-342: “For this reason, we used and opening thirty times larger in area than the 40mm2 described in [2]”. It should be “an” instead of “and”.

line 363: This sentence is poorly worded: Though the statistical model analyzing these proportions alone gives no clue of which process might be causing the observed results, we favor the second option because our previous results indicate that all cages have a similar probability of 366 flies dispersing (table 1; no effect of release cage).

Please rephrase this sentence to something along the lines of: While there is no statistical test/evidence in this study to support a greater influence of one process over the other (i.e. immigration vs. emigration), given that our results indicate all cages have a similar probability of flies dispersing, an observed proportion increase is likely due to higher immigration

Reviewer #3: Thank you for your revisions, they appear to address most if not all concerns. The manuscript now reads much more clearly.

7. PLOS authors have the option to publish the peer review history of their article (what does this mean?). If published, this will include your full peer review and any attached files.

Reviewer #1: No

Reviewer #2: No

Reviewer #3: **Yes: **Sweta Agrawal

---

## [Author Response · Author response to Decision Letter 1]

30 Sep 2020

Response to reviewer and editor comments

PONE-D-20-15094R1

Phenotype-dependent habitat choice is too weak to cause assortative mating between Drosophila melanogaster strains differing in light sensitivity

PLOS ONE

Thank you for submitting your manuscript to PLOS ONE. After careful consideration, we feel that it has merit but does not fully meet PLOS ONE’s publication criteria as it currently stands. Therefore, we invite you to submit a revised version of the manuscript that addresses the points raised during the review process.

The manuscript is greatly improved. I just have a couple of minor suggestions.

I also recommend that you read over the language carefully, as there might have been some other bloopers that I missed. Mind in particular the figure legends and table headings.

Editor

Minor

Comment: “This is because the white-eye mutation is a pleiotropic trait with a number of mildly impairing effects on the ...”

Actually the mutation is not a trait! Rephrase, and explain which traits are affected by mutations in white?

Response: Rephrased to: “This is because the mutation responsible for the white-eye phenotype also has an additional number of mildly impairing effects on the flies (see discussion)”, as we elaborate on the effects of the white mutation in the discussion. 

C: delta AIC in table - define as a footnote.

Defined.

C: “(heading level 2)”

R: We are not sure how this should be indicated. Currently, the heading levels in the manuscript are indicated by font size only: 18pt. for section titles and 16pt. for sub-sections.

Line 268

C: “Solid lines represent TWO replicates”

R: Changed to “represent the two replicates for each treatment”.

Line 280

C: Reword. “only explanatory variable fits the worst (Table 1)”

R: Changed to “the one using release cage as the only explanatory variable has the worst fit”.

C: Defined “SEM”

R: Changed all instances for the spelled-out version “Standard error of the mean”

Line 319

C: Reword “and very different form one where” and “choice is maximally phenotype-dependent”

R: Rephrased paragraph to “would be much closer to a scenario with random habitat choice than one where settlement choice is totally phenotype-dependent”

Table 3.

C: Make title more transparent

“Assortative mating scores for each simulation.”

R: Changed to “Proportion of same-phenotype pairs for each simulation”.

Line 337

C: Expand, names of authors “(personal observation)”

R: Changed.

Line 342

Still poor wording “We mitigate this effect statistically by...”

R: now reads: “In order to correct for this effect, we compare the results across all possible release cages”. 

Reviewer 1:

 (No Response)

Reviewer 2:

Reviewer #2: The authors have done a good job in cleaning up language and providing clarifications. Here are two comments that need to be addressed:

C: typo on line 341-342: “For this reason, we used and opening thirty times larger in area than the 40mm2 described in [2]”. It should be “an” instead of “and”.

R: Corrected.

C: line 363: This sentence is poorly worded: Though the statistical model analyzing these proportions alone gives no clue of which process might be causing the observed results, we favor the second option because our previous results indicate that all cages have a similar probability of 366 flies dispersing (table 1; no effect of release cage).

Please rephrase this sentence to something along the lines of: While there is no statistical test/evidence in this study to support a greater influence of one process over the other (i.e. immigration vs. emigration), given that our results indicate all cages have a similar probability of flies dispersing, an observed proportion increase is likely due to higher immigration

E: Reworded to “Although there is no evidence from the statistical models alone that supports a stronger influence of one process over the other, given that our previous results indicate that all cages have a similar probability of flies dispersing (table 1; no effect of release cage), an increase in the proportion of dispersers in a cage is likely due to higher immigration”.

Reviewer 3:

Reviewer #3: Thank you for your revisions, they appear to address most if not all concerns. The manuscript now reads much more clearly.

We thank the editor and all three reviewers for their kind and useful comments.

---

## [Editor Report · Decision Letter 2]

2 Oct 2020

Phenotype-dependent habitat choice is too weak to cause assortative mating between Drosophila melanogaster strains differing in light sensitivity

PONE-D-20-15094R2

Dear Dr. Peralta-Rincón,

We’re pleased to inform you that your manuscript has been judged scientifically suitable for publication and will be formally accepted for publication once it meets all outstanding technical requirements.

Kind regards,

Arnar Palsson, Ph.D.

Academic Editor

PLOS ONE
---

## [Editor Report · Acceptance letter]

6 Oct 2020

PONE-D-20-15094R2 

Phenotype-dependent habitat choice is too weak to cause assortative mating between *Drosophila melanogaster* strains differing in light sensitivity 

Dear Dr. Peralta-Rincón:

I'm pleased to inform you that your manuscript has been deemed suitable for publication in PLOS ONE. Congratulations! Your manuscript is now with our production department. 

Kind regards, 

on behalf of

Dr. Arnar Palsson 

Academic Editor

PLOS ONE